# Biocidal Coatings from Complexes of Carboxylated Latex Particles and a Linear Cationic Polymer

**DOI:** 10.3390/polym14214598

**Published:** 2022-10-29

**Authors:** Irina G. Panova, Evgeniya A. Shevaleva, Inessa A. Gritskova, Nataliya G. Loiko, Yury A. Nikolaev, Olga A. Novoskoltseva, Alexander A. Yaroslavov

**Affiliations:** 1Faculty of Chemistry, Lomonosov Moscow State University, 119991 Moscow, Russia; 2MIREA—Russian Technological University, 119571 Moscow, Russia; 3Department of Microbiology, Federal Research Center “Fundamentals of Biotechnology” RAS, 119071 Moscow, Russia

**Keywords:** carboxylated butadiene–styrene latex, biocidal polycation, interpolyelectrolyte complex, antimicrobial film

## Abstract

A linear polycation, poly(diallyldimethylammonium chloride), electrostatically interacts with anionic latex particles from a carboxylated butadiene–styrene copolymer in aqueous solution thus forming an interpolyelectrolyte complex. A mutual neutralization of oppositely charged latex and polycation groups occurs at *W* = latex/polycation = 50 *w/w* ratio. At *W* = 27, an ultimate polycation adsorption is reached, resulting in the formation of positive polycomplex particles, while at *W* ˂ 27, two-component systems are formed composed of positive polycomplex particles and free polycation. A film created from the *W* = 12 formulation shows a high toxicity to Gram-positive and Gram-negative bacteria and yeast. Repeated washing the film leads to partial removal of polycation and a 50% decrease in the activity of the film only towards Gram-negative *Pseudomonas aeruginosa*. The results indicate the potential for use of the mixed polymer formulations for the fabrication of antimicrobial films and coatings.

## 1. Introduction

Aqueous suspensions of polymer microspheres (polymer latexes) are widely applied in manufacturing protective coatings and binding (“gluing”) dispersed particles [1,2,3], due to the ability of latex particles to adhere to various surfaces and form highly elastic and strong films. These techniques find use in the paint/textile/paper industries, the production of food packaging, construction and road works, etc. [4,5,6,7,8]. Among others, butadiene/styrene binary latex (BSL) stands out with a 40% share of the world’s synthetic elastomer production [9]. In the molecular structure of BSL particles, flexible butadiene chains and rigid styrene chains are combined to give the latex coatings good mechanical properties, such as being waterproof and heat and abrasion resistant [10,11,12]. Incorporation of carboxylic groups in BSL increases the colloidal stability of latex particles in an aqueous environment and ensures the reactivity of the latex particles and latex coatings [13,14,15].

At the same time, the surfaces are able to accumulate pathogenic microorganisms, thus inducing infectious diseases [16,17,18,19]. In order to minimize the chance of infection, the surfaces are modified to give them antibacterial properties [20,21]. This can be accomplished, for example, via the addition of conventional low molecular weight bioactive additives, such as zinc oxide or silver particles, to latex formulations [22,23,24]. Additionally, a bactericidal effect can be expected after the mixing of latex particles with cationic polymers (polycations) known for their antimicrobial properties [25,26]. The latter is owing to an irreversible interaction between the positive groups of polycations and the negative surfaces of microorganisms. Such interaction inhibits functioning of the surface cell receptors and disrupts the cell membrane organization, thereby resulting in the death of cells [27,28,29].

In the current article, we describe aqueous latex formulations with commercial carboxylated (negatively charged) BSL microspheres that are 100 nm in diameter (cBSL). For bactericidal activity, the microspheres were modified with a cationic polymer, poly(diallyldimethylammonium chloride) (PDADMAC). We analyzed the polycation-to-latex microsphere complexation and the stability of the resulting polymer–latex complexes against aggregation, prepared films (coatings) from the aqueous formulations, and examined the bactericidal properties of the coatings. The results allow for the correlation between the composition and bactericidal activity of mixed latex–polycation formulations, and the optimization of antimicrobial properties of latex–polycation coatings. 

## 2. Materials and Methods

### 2.1. Materials

The materials used included a 50 wt% carboxylated butadiene–styrene latex (cBSL) with polymer microspheres consisting of a butadiene/methylstyrene/methacrylic acid ternary copolymer (70/30/2; residual styrene content of 0.01 wt%; JSC Voronezhsintezkauchuk, Russia), as well as poly(diallyldimethylammonium chloride) (PDADMAC) with *Mw* = 200–350 kDa, poly(sodium-4-styrenesulfonate) (PSS) with *Mw* ~ 70 kDa, hydrochloric acid, sodium hydroxide, and sodium dihydrogen phosphate monohydrate (all from Sigma-Aldrich, St. Louis, MO, USA). 

Interpolyelectrolyte complexes (IPEC) were prepared by mixing cBSL and PDADMAC solutions with corresponding concentrations. The concentrations of the polymers were given in the molar concentration of their ionic units, i.e., their carboxyl [COOH] and quaternary amino [N] groups. 

A quantity of 7 mL of a 4 wt% latex formulation or 4 wt% latex-PDADMAC formulation was deposited over a 26.4 cm^2^ cellophane substrate and dried at 80 °C to constant weight. The dried film (0.2 mm in thickness) was removed from the substrate, then placed in a Petri dish and covered with 15 mL of bi-distilled water. After 15 min, the water was removed, and the film was dried again at 80 °C to constant weight. The washing/drying procedure (the “film wash-off”) was repeated up to 7 times.

### 2.2. Methods

The pH values of solutions were assessed using a Corning 340 pH meter (USA) equipped with a combination pH glass electrode with integrated temperature sensor.

A conductometric titration of the latex suspension was conducted with a CDM 83 conductivity meter with a platinum electrode PP1042 (Radiometer, Denmark). 

Mean hydrodynamic diameter of particles was determined by dynamic light scattering at the fixed scattering angle (90°) in a thermostatic cell with a Brookhaven Zeta Plus instrument (USA). Software provided by the manufacturer was employed to calculate diameter values. Electrophoretic mobility (EPM) of particles was measured by laser microelectrophoresis in a thermostatic cell using a Brookhaven Zeta Plus instrument with the corresponding software.

Visualization of particles with transmission electron microscopy was performed with a JEM-2000 FXII unit (JEOL, Japan).

The antimicrobial properties of aqueous polymer formulations were quantified via determination of their minimum inhibitory concentrations (MIC) and minimum bactericidal concentrations (MBC). Three types of microorganisms (MOs) were used: Gram-negative bacteria *Pseudomonas aeruginosa* 4.8.1, Gram-positive bacteria *Staphylococcus aureus* 209P, and yeast (eukaryotes) *Yarrowia lipolytica* 367-2 (MOs collections of Research Center of Biotechnology RAS). The procedure was as follows: M9 medium was poured into 18 mL glass test tubes (2 mL each) and the M9 medium composition (pH 7.0) was as follows (g L^−1^): Na_2_HPO_4_—6; KH_2_PO_4_—3; NaCl—0.5; NH_4_Cl—0.2; MnSO_4_—0.0004; MgSO_4_—0.0025; CaCl_2_—0.0002; and glucose—10 (all from Sigma-Aldrich, USA). Then, various aliquots of 4 wt% polymer solutions were added, so that the polymer concentration ranged from 0 to 2 wt%. The tubes were inoculated (1 vol%) with the bacteria *P. aeruginosa* or *S. aureus,* or the yeast *Y. lipolytica*, using 1-day cultures of the stationary growth phase, and were placed on a shaker at 28 °C. After 2 days, the growth of MOs was assessed. The lowest polymer concentration at which no growth of the test cultures was observed visually was taken as MIC. Then, aliquots of the cultures from the tubes with polymer concentrations ≥ MIC were plated on Petri dishes with solid Miller’s Luria–Bertani (LB) broth medium (VWR, USA) supplemented with agar (3% Bacteriological Agar, Helicon, Russia). After incubation of the Petri dishes for 2 days at 28 °C, the growth of the MOs on a dense medium was evaluated. MBC was defined as the lowest concentration of the polymer at which there was no MO growth [30,31,32].

In order to study the antimicrobial activity of latex and mixed latex–polymer films, five types of MOs were used: Gram-negative bacteria *P. aeruginosa* 4.8.1 and *Escherichia coli* K12, Gram-positive bacteria *S. aureus* 209P and *Micrococcus luteus* NCIMB 13267, and yeast (eukaryotes) *Y. lipolytica* 367-2 (MOs collections of Research Center of Biotechnology RAS). The microorganisms were grown in 250 mL flasks with cotton plugs, which contained 50 mL of the LB broth medium, and were stirred at 120 rpm for one day at 28 °C. After that, the cell cultures were diluted with sterile distilled water so that 50 µL of the prepared suspension contained from 10^2^ to 3 × 10^4^ cells (depending on the specific experiment). 

A 1 cm^2^ latex or mixed latex–polymer film was placed on a sterile coverslip, and 50 μL of the diluted cell suspension was deposited on top. The sample was incubated for a period of time ranging from 5 to 30 min; then the cell suspension was washed out of the film into a Petri dish with the LB agar medium using 300 µL of sterile distilled water. The Petri dish was incubated for 2–3 days at 28 °C followed by counting the grown colonies of MOs. 

The inhibitory effect of the latex and mixed latex–polymer films was calculated as a percentage of the control (the initial number of cells). Biological experiments were carried out in triplicate. Statistical processing of the experimental data was carried out using Excel, with estimation of the arithmetic mean and standard deviation. The significance of differences between the variants was evaluated using Student’s *t*-test, with *p* < 0.05 considered significant.

## 3. Results and Discussion

### 3.1. Colloid-Chemical Properties of Latex

Analysis of 0.01 wt% aqueous cBSL suspension with dynamic light scattering gave a narrow size distribution of polymeric microspheres with an average hydrodynamic diameter of 95 nm (Figure 1a). The monodispersity of microspheres was confirmed by transmission electron microscopy which showed one type of spherical particle with an average diameter of 98 nm (Figure 1b).

Carboxylic groups of methacrylic acid rendered a negative charge to the cBSL microspheres; their electrophoretic mobility or EPM (a parameter associated with the microsphere surface charge) was −4.5 (µm/s)/(V/cm), which suggested excellent stability of the cBSL microspheres against aggregation in an aqueous solution.

The quantity of carboxylic groups on the surface of cBSL microspheres was measured with reverse conductance titration. Before titration, NaOH was added to the latex until a pH value of 11 was achieved, when all cBSL carboxylic groups were converted into salt form.

The conductance titration curve (Figure 2) consisted of three sections, which reflected the titration of an excess alkali (1), carboxylic groups (2), and the accumulation of free HCl in solution (3). From the experimental data, the content of carboxylic groups was found to be equal to 0.06 moles per 1 g of cBSL.

### 3.2. Formation of the Complexes between cBSL Microspheres and PDADMAC

Addition of a cationic PDADMAC aqueous solution to an anionic cBSL suspension altered the EPM of latex particles as shown in Figure 3a; here *Z* = [N^+^]/[COOH] and represents the molar ratio of the concentration of PDADMAC quaternary amino groups to the concentration of cBSL carboxylic groups. Addition of polycation first led to neutralization of the cBSL microsphere charge; at higher polycation concentration, the microspheres acquired a positive charge. These changes in the EPM value were evidence of electrostatic complexation of PDADMAC with cBSL microspheres.

The cationic PDADMAC contains quaternary amino groups, which give the maximum positive charge to a PDADMAC macromolecule at pH 7, which was the experimental condition. In other words, all the cationic PDADMAC groups participated in the electrostatic complexation with the cBSL microspheres. At EPM = 0, the concentration of the positive PDADMAC groups involved in the complexation ([N^+^]_EPM=0_) was equal to the concentration of the negative (ionized) cBSL groups ([COO^−^]_EPM=0_). This allowed, based on the data in Figure 3a, the estimation of the maximum degree of carboxylic cBSL groups electrostatically complexed with PDADMAC: *ω* = [COO^−^]_EPM=0_/[COOH] = [N^+^]_EPM=0_/[COOH] = 0.82.

In the literature [33], the degree of dissociation of the carboxylic groups at pH 7 was found to be *α* ≈ 0.7. This means that all ionized COO^−^ groups of the cBSL complexed electrostatically with the PDADMAC quaternary amino groups in a pH 7 solution and formed interpolyelectrolyte complexes (IPEC). Extra ionized COO^−^ groups (*ω* ≈ 0.1) appeared due to a cooperative displacement of protons from carboxylic groups by the interacting PDADMAC [34]. Residual 1 − 0.82 = 0.18 carboxylic cBSL groups did not form ionic bridges with cationic PDADMAC. The *Z* = [N^+^]_EPM=0_/[COOH] = 0.82 ratio thus corresponds to the electroneutral (saturated) IPEC in a pH 7 solution. On the other hand, the “zero charge” point can be expressed as a ratio between the total weight concentration of cBSL and the weight concentration of PDADMAC at EPM = 0: *W* = (cBSL)_wtcon_, _total_/(PDADMAC)_wtcon_, _EPM=0_ = 50. The variable *W*, expressing the weight ratio of components, is convenient as it highlights the role of latex as a basis of mixed formulations, and it facilitates operating with integers. A decrease in the *W* value below 50, or an increase in the *Z* value over 0.82, was accompanied by the appearance of a positive charge on the surface of cBSL microspheres; the maximum EPM value = +2 (μm/s)/(V/cm) was achieved at *W* = 27 (*Z* = 1.5). It was shown previously [14,35,36] that the ultimate positive EPM value corresponded to the maximum adsorption of a cationic polymer onto the surface of negative colloid particles. After the EPM reached the ultimate positive value, the binding of PDADMAC stopped.

The latex-to-polycation complexation affected the size of particles in the system as was shown with dynamic light scattering (Figure 3b). Neutralization of the microsphere surface charge as a result of interacting with the polycation caused the particles to aggregate; in the EPM range between −1 (μm/s)/(V/cm) and +1 (μm/s)/(V/cm), phase separation was observed. In an excess of PDADMAC, at *W* < 27 (or Z > 1.5), the size of latex–polycation complex particles decreased to 170 nm. In this PDADMAC concentration range, the stability of complex particles against aggregation was due to the high positive charge brought by the adsorbed polycation.

### 3.3. Investigation of Antimicrobial Properties of Polymer–Latex Formulations in Aqueous Solution

The first step was to examine the antimicrobial properties of polymer formulations in aqueous solution. Using standard methods (see Section 2) the polymer concentrations were found which caused inhibition of microorganism growth and their death, MIC and MBC respectively. The results for the five polymer formulations, i.e., anionic cBSL, cationic PDADMAC, and three cBSL/PDADMAC polycomplexes with different component ratios, are summarized in Table 1. Anionic cBSL showed a rather low antibacterial effect: the latex formulation did not suppress the growth of the Gram-negative bacteria *P. aeruginosa* even at the highest tested latex concentration of 2 wt%. For the Gram-positive bacteria *S. aureus* and yeast *Y. lipolytica*, their growth was suppressed at 0.7 and 1.5 wt% cBSL concentrations, respectively. Slightly higher cBSL concentrations caused the death of the microorganisms.

The toxicity of cBSL/PDADMAC polycomplexes depended on their composition. The negative IPEC with *W* = 40, in which all PDADMAC was bound to the cBSL, showed toxicity comparable with that for the initial cBSL. Both positive IPECs with *W* = 12 and 8 demonstrated a higher toxicity level, which increased as the content of PDADMAC in the polycomplex increased. Recall that the maximum content of PDADMAC was achieved at *W* = 27. At the higher *W* value, some PDADMAC remained unbound to the latex microspheres; it is the unbound cationic polymer that contributed to the toxicity of the polymer–latex formulations with *W* = 12 and 8. The individual cationic PDADMAC was the most toxic: it suppressed the bacteria/yeast growth at 0.0005–0.001 wt% and caused their death at 0.001–0.002 wt%, which was in agreement with earlier published results [37,38].

### 3.4. Antimicrobial Activity of cBSL and W = 12 cBSL/PDADMAC Films

Antimicrobial activity of polymer formulations in the films is reflected in Table 2 as the percentage of surviving cells after contact with the films. The experiments were carried out with films prepared from the *W* = 12 formulation; the surviving cells were quantified 5, 15, and 30 min after their deposition onto the films. The results were compared with those for the cBSL films devoid of polycation.

The survival of the cells on the cBSL films was found to be rather high and changed little over time. The percentage of surviving cells was 71–82 wt% for the Gram-negative bacteria (columns 6 and 8) and 72–97 wt% for the Gram-positive bacteria (columns 2 and 4). The yeast showed a lower result: 37–47 wt% (column 10). Deposition onto the mixed cBSL/polycation films was accompanied by 100% death of the Gram-positive bacteria (columns 3 and 5) and the yeast (column 11) within 5 min. The Gram-negative bacteria *E. coli* were guaranteed to die within 15 min (column 7). The Gram-negative bacteria *P. aeruginosa* showed the greatest resistance: 8% was still alive 30 min after deposition onto the cBSL/polycation films (column 9).

### 3.5. Influence of Initial Titer of Deposited Microorganisms on the Survival of the Gram-Negative Bacteria P. aeruginosa

Additionally, we examined how the density of bacterial contamination affected the survival of deposited microorganisms. In this experiment, the Gram-negative bacteria *P. aeruginosa* were used, which demonstrated the maximum survival on the bactericidal cBSL/polycation film. The results are summarized in Table 3. An increase in the density of bacterial contamination had a slight effect on the bactericidal activity of the cBSL films. As the number of deposited cells increased by more than an order of magnitude, the percentage of surviving cells remained at 75–85% for a 5 min exposure and 66–71% for a 30 min exposure (columns 2, 4, 6, 8).

A different situation was observed when the cells were added to the mixed *W* = 12 cBSL/polycation films. In this case (columns 3, 5, 7, 9), the percentage of surviving cells was lower in comparison with that for the cBSL film without cationic polymer. The 30 min exposure was sufficient for deactivation of the majority of the deposited cells. 

### 3.6. Biocidal Properties of W = 12 cBSL/PDADMAC Films after Washing with Water

In the biocidal *W* = 12 films, the amount of cationic PDADMAC exceeded that required for latex saturation. The polycation could be removed from the films when treated with water. This situation occurs when the films are operated in a humid atmosphere or washed with water to remove contaminants. 

In order to quantify the amount of PDADMAC removed from the films, the *W* = 12 latex–polycation film was placed in bi-distilled water for 15 min, then taken out and dried to constant weight. The washing/drying procedure (the “film washing”) was repeated up to seven times. The initial weight of the control latex film deprived of PDADMAC was 279 ± 1.5 mg. After the first washing, the weight of the latex film decreased by 1 mg, which was within experimental error, and did not change in the course of subsequent washings. This result demonstrated that there was no removal of colloidal particles from the latex film. Additionally, the wash water was analyzed for the presence of latex particles using dynamic light scattering, and no signal was detected. This was in agreement with the data from gravimetric analysis.

As for the *W* = 12 latex–polycation film with a weight of 279 ± 1.5 mg, it lost 9 mg in the course of the first washing and 4 mg in the course of the second. Subsequent washings extracted nothing from the latex–polycation film. Taking into account the 100% stability of the latex film to re-watering, one can conclude that only the polycation was removed from the mixed latex–polycation film in the washing/drying experiments. This conclusion was confirmed by the data from the following experiment: The wash water, 30 mL in total, was collected in a glass bottle, then 3 mL was taken and titrated by an aqueous solution of an anionic polymer, PSS. It has been shown that anionic polymers, including PSS, bind electrostatically to cationic polymers thus producing products known as interpolyelectrolyte complexes (IPEC) which are stabilized by multiple salt bonds between the oppositely charged units of both components [34]. In our experiment, a progressive increase in the volume of PSS solution resulted in the appearance of particles in the titrated system whose charge became decreasingly positive, then reached EPM = 0 point, and finally became negative (Figure 4, curve 1). In parallel, the size of the particles in the titrated system was measured (Figure 4, curve 2), which showed first a rise and then a decrease with the maximum at a PSS content corresponding to EPM = 0. These plots indicate the interaction of an anionic PSS with a cationic object [39], which could only be cationic PDADMAC removed from the film with water.

The content of PDADMAC in the initial latex–polycation film was 21 mg and after two washings decreased to 21 − (9 + 4) mg = 8 mg. Thus, (8 mg/21 mg) × 100% = 38% of PDADMAC was retained in the latex–polycation film after seven intensive washings, while the extraction of polycation was completed after the second washing/drying cycle. 

The antimicrobial properties of pre-washed latex–polycation films were tested towards all five microorganisms mentioned above: two Gram-positive—*S. aureus* and *M. luteus*, two Gram-negative—*E. coli* and *P. aeruginosa*, and yeast *Y. lipolytica*. The films were prepared from the *W* = 12 formulation, which lost 62% of the polycation in the course of the first two washing procedures. 

As shown in Table 2, the Gram-positive bacteria and the yeast were completely deactivated 5 min after their deposition onto the *W* = 12 latex–polycation film (columns 3, 5, 11). The same was observed when the Gram-positive bacteria and the yeast were deposited onto the films pre-washed with water one time and seven times (data not shown). In other words, the loss of 62% of PDADMAC had no effect on the antimicrobial activity of the latex–polycation films towards the Gram-positive bacteria and the yeast. 

Different results were obtained with the Gram-negative bacteria. The washing procedure significantly reduced the antimicrobial activity of the latex–polycation films towards *P. aeruginosa* (Table 4, columns 2–4). After seven washing/drying cycles, the films were able to deactivate only 56% of the deposited bacteria, which is not important from a practical point of view. In contrast, the seven-fold washed films retained a high activity towards *E. coli* and killed 99.9% of cells within 30 min after deposition (Table 4, columns 5–7). The above data show the relationship between the composition of the washed latex–polycation films and their antimicrobial activity. Recall that the electroneutral latex–polycation complex with EPM = 0 is formed at the latex-to-polycation wt/wt ratio *W* = 50. The value of *W* = 27 corresponds to the saturated positively charged complex with maximum amount of adsorbed polycation. In the washing/drying experiments, the film with *W* = 12 was initially taken, which, after seven intensive washings, lost 62% of the polycation. This corresponded to the film with *W* ≈ 30. Thus, seven washings resulted in a quantitative removal of the polycation unbound to the latex particles, and formation of the films from the saturated positively charged complex. It follows from this that the initial and washed films both produced a maximum positive charge on the outer film surface. 

This result is in good agreement with the result found of no effect of the washing/drying procedure on the antimicrobial activity of the latex–polycation films towards most bacteria and the yeast. The only exception was Gram-negative *P. aeruginosa* bacteria, which demonstrated higher viability on the washed film in comparison with the initial unwashed film (Table 4, columns 2, 3, and 4). Currently, we do not have a satisfactory explanation for this phenomenon. It could be due to the specific structure of the *P. aeruginosa* bacterial wall, or it may result from a higher mobility of the polycation in the unwashed film, which gives it additional opportunities to attack the cells. In any event, this observation requires additional experiments.

## 4. Conclusions

Electrostatic binding of cationic PDADMAC to anionic cBSL microspheres in an aqueous solution resulted in the formation of cBSL/PDADMAC complex particles, which achieved electro-neutrality at a cBSL/PDADMAC weight ratio *W* = 50. At *W* = 27, positively charged saturated complex particles were formed; at lower *W* values the solution contained the saturated complex and free polycation macromolecules. The mixed cBSL/PDDAMAC aqueous formulations with *W* ˂ 27 were stable against aggregation up to a 4 wt% total polymer concentration. 

Deposition of the cBSL formulation and mixed cBSL/polymer formulations onto a solid substrate followed by drying the samples produced cBSL and mixed cBSL/PDADMAC films. The cBSL films showed moderate stability to re-watering, and the mixed *W* = 12 films lost 62% of the polymer during seven repeating re-watering cycles. 

Aqueous solutions of cBSL and *W* = 40 formulations showed negligible toxicity towards Gram-positive bacteria, Gram-negative bacteria, and yeast. The oversaturated mixed formulations with *W* = 12 and 8, which contained an excess of free PDADMAC, demonstrated cytotoxicity comparable with that for the polycation alone. 

The mixed *W* = 12 films with an excess of PDADMAC, both the initial and seven times washed, killed 100% of Gram-positive bacteria and yeast within 5 min after their deposition. Gram-negative bacteria were more resistant: 30 min after deposition over the mixed *W* = 12 films, 99.9% of *E. coli* and 90% of *P. aeruginosa* died. The re-watering did not affect the activity of the mixed films to *E. coli* and significantly decreased the film activity to *P. aeruginosa*. The bactericidal activity of the mixed films decreased with an increasing density of deposited microorganisms. The results indicate a great potential for the use of mixed polymer formulations in the fabrication of antimicrobial films and coatings.

## Figures and Tables

**Figure 1 polymers-14-04598-f001:**
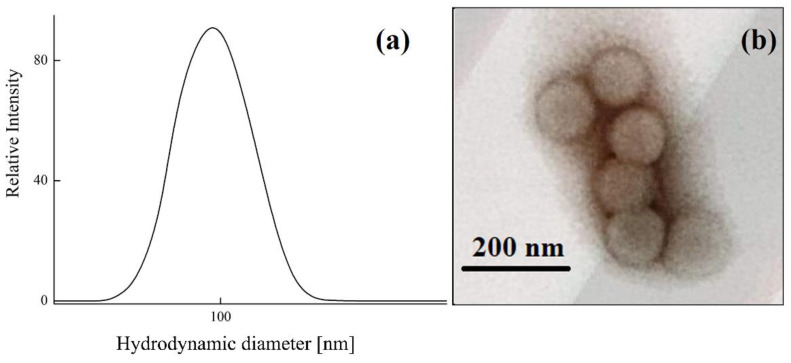
(**a**) Size distribution of cBSL latex particles in 10^−2^ M phosphate buffer aqueous solution with pH 7 at 25 °C. (**b**) Transmission electron microscopy image of cBSL microspheres.

**Figure 2 polymers-14-04598-f002:**
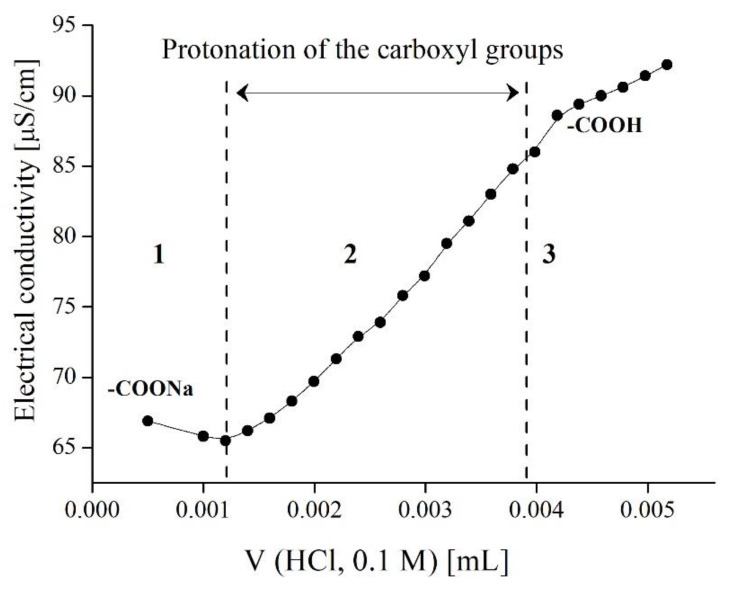
Conductometric titration curve of cBSL particles. Volume of the latex, 30 mL; mass of the cBSL particles, 0.05 g; temperature, 22 °C.

**Figure 3 polymers-14-04598-f003:**
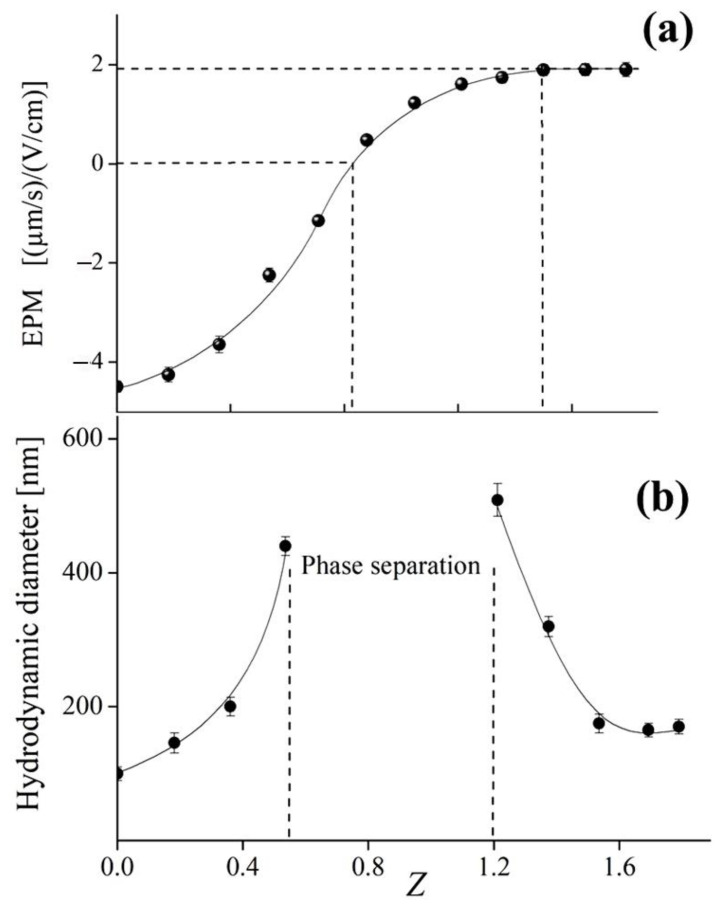
(**a**) Electrophoretic mobility and (**b**) hydrodynamic diameter of latex particles vs. molar ratio, *Z*. Concentration of latex carboxylic groups, 4 × 10^−5^ M; 10^−2^ M phosphate buffer aqueous solution with pH 7; temperature, 25 °C.

**Figure 4 polymers-14-04598-f004:**
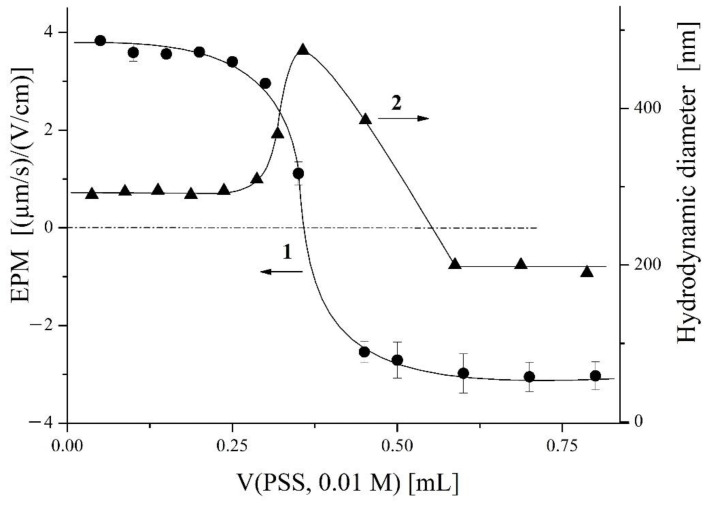
(1) Electrophoretic mobility and (2) hydrodynamic diameter of particles in the system vs. PSS concentration.

**Table 1 polymers-14-04598-t001:** Antimicrobial activity of polymer formulations in solution.

Formulation	Parameter	MIC and MBC Determination Results, wt%		
		*P. aeruginosa*	*S. aureus*	*Y. lipolytica*
cBSL	MIC	>2.0	0.7	1.5
	MBC	>2.0	1.7	>2.0
*W* = 40	MIC	>2.0	0.5	1.3
	MBC	>2.0	1.5	>2.0
*W* = 12	MIC	0.07	0.04	0.04
	MBC	0.09	0.04	0.06
*W* = 8	MIC	0.04	0.02	0.02
	MBC	0.075	0.02	0.04
PDADMAC	MIC	0.001	0.0005	0.0005
	MBC	0.002	0.001	0.0015

**Table 2 polymers-14-04598-t002:** Antimicrobial activity of cBSL (I) and *W* = 12 cBSL/PDADMAC films (II).

Exposure Time, min	Percentage of Survived Cells after Deposition onto Polymer Films									
	*S. aureus*		*M. luteus*		*E. coli*		*P. aeruginosa*		*Y. lipolytica*	
	I ^*)^	II ^**)^	I ^*)^	II ^**)^	I ^*)^	II ^**)^	I ^*)^	II ^**)^	I ^*)^	II ^**)^
**1**	**2**	**3**	**4**	**5**	**6**	**7**	**8**	**9**	**10**	**11**
5	97.7 ± 3.6	0	83.1 ± 2.1	0	82.0 ± 2.5	2.1 ± 0.1	77.1 ± 2.4	40.6 ± 1.9	47.6 ± 1.3	0
15	96.4 ± 3.1	0	77.6 ± 1.6	0	80.0 ± 3.4	0	74.4 ± 2.2	15.0 ± 0.8	43.1 ± 1.7	0
30	93.7 ± 4.8	0	72.0 ± 3.0	0	73.3 ± 4.9	0	71.1 ± 4.6	8.3 ± 0.6	37.9 ± 2.4	0

^*)^ cBSL film. ^**)^ cBSL/PDADMAC film with *W* = 12.

**Table 3 polymers-14-04598-t003:** Percentage of survived cells after deposition onto polymer films vs. *P. aeruginosa* contamination.

Exposure Time, min	Number of Cells Per Film							
	125 ± 7		550 ± 54		700 ± 65		1500 ± 123	
	I ^*)^	II ^**)^	I ^*)^	II ^**)^	I ^*)^	II ^**)^	I ^*)^	II ^**)^
**1**	**2**	**3**	**4**	**5**	**6**	**7**	**8**	**9**
5	85.6 ± 3.5	23.0 ± 1.5	79.8 ± 3.1	32.1 ± 1.2	77.1 ± 2.4	40.6 ± 1.9	76.3 ± 2.8	63.3 ± 2.5
15	73.6 ± 3.2	6.5 ± 0.4	74.2 ± 2.9	10.7 ± 0.7	74.4 ± 2.2	15.0 ± 0.8	71.7 ± 3.9	31.7 ± 1.8
30	69.6 ± 3.6	1.6 ± 0.1	66.8 ± 4.3	5.8 ± 0.6	71.1 ± 4.6	8.3 ± 0.6	68.2 ± 4.0	12.1 ± 1.1

^*)^ cBSL film. ^**)^ cBSL/PDADMAC film with *W* = 12.

**Table 4 polymers-14-04598-t004:** Influence of washing *W* = 12 cBSL/PDADMAC films on the survival of Gram-negative bacteria *P. aeruginosa* and *E. coli* (1500 ± 123 cells per each film).

Exposure Time, min	Percentage of Survived Cells after Deposition onto Polymer Films					
	*P. aeruginosa*			*E. coli*		
	Number of the Washing/Drying Cycles			Number of the Washing/Drying Cycles		
	No	1	7	No	1	7
**1**	**2**	**3**	**4**	**5**	**6**	**7**
5	63.3 ± 2.5	77.4 ± 3.1	78.9 ± 3.5	5.2 ± 0.4	8.3 ± 3.0	9.8 ± 0.8
15	31.7 ± 1.8	58.2 ± 3.0	60.1 ± 4.2	0.9 ± 0.07	1.5 ± 3.0	2.1 ± 0.06
30	12.1 ± 1.1	40.3 ± 2.6	44.3 ± 2.9	0.08 ± 0.006	0.1 ± 3.0	0.1 ± 0.009

## Data Availability

Not applicable.

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
