# Peer review of "Biocidal Coatings from Complexes of Carboxylated Latex Particles and a Linear Cationic Polymer"

_polymers, 2022, doi:10.3390/polym14214598_

Round 1

Reviewer 1 Report

The paper summarizes the action of antibacterial polymers both in solution and as coatings. A linear polycation, poly(diallyldimethylammonium chloride), electrostatically interacts with anionic latex particles from carboxylated butadiene-styrene copolymer in aqueous solution for fabrication of antimicrobial films and coatings. The results indicate a great potential of the mixed polymer formulations for fabrication of antimicrobial films and coatings.

The quality of the flow of arguments is quite well. However, there are some unclear points and/or more detailed discussion is necessary before the publication.

Comment 1

Regarding in solution phase activity, is there any relationship between aggregation (complex) values of the interpolyelectrolyte complexes (IPEC) with the biocidal activity reported in Table 1? Also, MBC for PolyDADMAC is missing in Table 1.

Comment 2

Author briefly explained the antimicrobial properties of polymer formulations in aqueous solution and surface. The size and charge of the particles in solution was investigated by DLS and electrophoretic mobility, respectively. However, solution and surface behavior (biocidal activity) of the interpolyelectrolyte complexes (IPEC) could differ. Surface analysis of the IPEC (such as cationic charge density using NaFlourescein method and surface tension via contact angle etc) are missing to speculate about the structure- property relationship at solid surface.

Comment 3

Author claim that [Line 272-285] “Taking into account the 100% stability of the latex film to re-watering, one can conclude that only the polycation was removed from the mixed latex-polycation film in the washing/drying experiments.” Based on that point polycation was 38% of PDADMAC retained in the latex-polycation film after 7 intensive washings, while the extraction of polycation was completed after the second washing/drying cycle.” What is the evidence for the PDADMAC released from the surface? Solution phase (after washing) should be evaluated and characterized using instrumental techniques such as SEC GPC, elemental analysis, FTIR, NMR to prove that the leached out sample is polycation.

Comment 4

“Section 3.6. Biocidal properties of W = 12 cBSL/PDADMAC films after washing with water” It was observed that repeated washing of the film leads to partial removal of polycation and a 50% decrease in the activity of the film only towards Gram-negative Pseudomonas aeruginosa. Why? What is the reason? Is there any difference between biofilm formations of each microorganisms on the surface? Discussion on killing mechanism can be provided to understand the surface properties. After washing the surface the cationic charge density, hydrophobicity can be evaluated to speculate about the mechanism. Surface properties of the film after deposition of microorganisms can also be evaluated using AFM, confocal microscopy etc. to further understand the surface properties of the film. This part is also relevant with Comment 2.

Specific comments

Table 4….. “Gramm-negative” should be revised.

Author Response

Comment 1

Answer: We made corresponding corrections in Table 1.

We used in the research the anionic IPEC with W=40 as well as the mixtures of cationic IPEC and PDADMAC with W=12 and 8 whose sizes were within 150-200 nm and did not change in time. So, we did not register a relationship between aggregation of IPECs and their biocidal activity.

Comment 2

Answer: The non-specific biocidal activity of polycations is known to be due to the huge positive charge brought in by cationic groups of polymers. Interaction of the cationic polymer groups with the negative surface of microorganisms inhibits functioning of the cell receptors and disrupts the cell membrane organization thereby resulting in the death of cells. This assertion is supported by references [27-29] in our article.

For this reason, we concentrated in our work on the study of composition of the latex-polycation complex, in which it is the polycation that contributed to the total biocidal activity. Aqueous solutions of the latex-polycation complexes were deposited over the glass plate and dried that resulted in formation of the coatings whose compositions in terms of later-to-polycation ratio were equal to the composition of the initial latex-polycation complexes. We showed that repeated washing procedures were accompanied by a removal of the polycation from the coatings. In parallel, we registered a decrease in the biocidal activity of coatings. So, a quantitative relationship between the complex composition and biocidity was demonstrated.

Comment 3

Answer: The coatings consisted of only two components: latex particles and polycation macromolecules. We showed in a separate experiment that latex was not removed from the coatings in the course of the repeating washing/drying cycles. It means definitely that it is the polycation that was removed from the coatings with water.

We also carried out additional experiments for detection of latex and polycation in the washing water. First, the control latex film was washed with water as described in the article, and the washing water was tested with the use of dynamic light scattering and electrophoresis. No colloidal particles was detected. Second, the mixed latex-polycation film was washed with water and the washing water was tested again with the same methods. In the two first washing waters, particles with positive charges were detected, which were obviously attributed to the polycation macromolecules. Further watering procedures did not show detectible particles.

Comment 4

Answer: First, we did not examine the formation of microorganism biofilms over the latex-polycation coatings. This was outside the scope of our article.

Second, we did not analyzed the mechanism of the polycation biocidal effect. We adopted the conventional scheme according to which the biocidal activity of the latex-polycation coatings is associated with the cationic groups of the embedded polycation. The less polycation, the lower biocidal activity. It is this correlation that was observed in our experiments.

Third, the coatings consisted of only two components: latex particles and polycation macromolecules. We showed in a separate experiment that latex was not removed from the coatings in the course of the repeating washing/drying cycles. It means definitely that it is the polycation that was removed from the coatings with water.

 Specific comments

Answer: The misprint was corrected.

Reviewer 2 Report

Bacterial associated infections that can damage human’s health are very serious issues worldwide. Coating substrates with antibacterial materials has been recognized as an effective method to inhibit bacterial colonization.

In this work, the authors prepared a series of interpolyelectrolyte complexs by using commercial carboxylated butadiene/styrene binary latex (cBSL) and cationic poly(diallyldimethylammonium chloride) (PDADMAC). It’s found that cBSL/PDADMAC polycomplexes in aqueous solution have high bactericidal activity. The antimicrobial activity of cBSL/W=12 cBSL/PDADMAC film was also investigated. Deposition onto the cBSL/polycation film (W=12) was accompanied by a 100% death of the Gram-positive bacteria and the yeast. The stability of this bactericidal coating showed that the W = 12 cBSL/PDADMAC film still has high bactericidal activity after washing with water several times.

This work provided a new kind of coating with promising bactericidal properties. But there is still some works to be done to meet the high standard requirements of this journal. I recommend the manuscript for publication after minor revision.

(1) Hemolysis test of the antibacterial coatings need to be done.

(2) Plate count method for bacterial enumeration is considered to be the gold standard in microbiology. The authors should provide the picture of the corresponding plates for Table 2 and 3.

(3) Some related references should be added, such as J. Mater. Chem. B, 2016,4, 1081-1089, Chin J Polym Sci, 2021,39, 1020–1028.

Author Response

(1)  Answer: This test concerns the destruction of red blood cells. We did not work with these cells. We worked with bacterial and yeast cells, which were deposited over the polymer-based coatings. It is these results which are described in our article. We see no reason to include red blood cells in the list of cells examined in our research.

(2) Answer: The plate count method is actually the gold standard in microbiology for bacterial enumeration. We are sure that raw data, such as photos of colonies, are not appropriate for a respected scientific journal like Polymers. Moreover, we do not keep such a photos, as we have hundreds of them. We usually keep basic data in digital/numeric form. This is a traditional experimental and journal practice.

(3) Answer: We added the recommended papers to the list of references.

Round 2

Reviewer 1 Report

The authors answered the questions, but I think some parts need to be revised

Related with Comment 3. Discussion/picture on the washing cycle and testing with the use of dynamic light scattering and electrophoresis for the mixed latex-polycation film is missing in the main manuscript.

Authors claim at Comment 4 that “The less polycation, the lower biocidal activity. It is this correlation that was observed in our experiments.” This should also be quantified with the surface potential and positive charge densities to understand structure-property relationship.

Author Response

1. Related with Comment 3. Discussion/picture on the washing cycle and testing with the use of dynamic light scattering and electrophoresis for the mixed latex-polycation film is missing in the main manuscript. 

Answer: Discussion of the washing procedure and latex/polycation detection, supplied by a corresponding picture, is now added to the text.

2. Authors claim at Comment 4 that “The less polycation, the lower biocidal activity. It is this correlation that was observed in our experiments.” This should also be quantified with the surface potential and positive charge densities to understand structure-property relationship.

Answer: In our article we examine the stability of the latex-polycation films in water surrounding (during the washing/drying cycles) and how a possible removal of polycation affects the biocidal activity of the films. That is we were interested in a formal connection between the removal and activity.

We examined neither the surface morphology nor even more so internal organization of the films. For this reason, we can say nothing about “structure” of the latex-polycation films and “structure-property relationship”.

What we can discuss is the composition of the films in terms of latex-to-polycation ratio, which is presented in the article as W. The W = 50 means the electroneutral latex-polycation complex with EPM = 0, the W = 27 the saturated positively charged complex with maximum amount of polycation on a single latex particle.

In the washing experiments, the film with W = 12 was taken, which after 7 intensive washings lost 60% of polycation. This corresponded to the film with W = 30. Thus, 7 intensive washings resulted in a quantitative removal of the most polycation, unbound to the latex particles, and formation of the films from the saturate positively charged complex. The initial W=12 film and the washed film both obviously produced a maximum positive charge on the outer film surface.

Now turn to Section 3.6 of the article. It is stated there that loss of polycation had no effect on the antimicrobial activity of the latex-polycation films towards Gram-positive bacteria and the yeast”. This is a good agreement with above observation.

The only exception is Gram-negative P. aeruginosa bacteria, which demonstrate higher stability being deposited both over the initial film and after 7-fold washing. The removal of polycation leads to a significant increase in their viability. Probably, it depends on specific structure of the bacterial wall. Or unbound polycation with higher mobility (in comparison with the bound to the latex) has additional opportunities to attack the cells. In any event, this observation requires additional experiments. Now we only preliminary results, which do not allow the reliable conclusions about the reasons for such behavior of the P. aeruginosa bacteria.

Corresponding additions are now made in the text.

Round 3

Reviewer 1 Report

The authors have made the necessary editing and improvement in the publication. However, I think it is necessary to quantify the charge density of the surface in particular. This study shows that the structure-property relationship is examined. The preparation of W = latex/polycation solutions and coatings at different ratios shows that the structure-property relationship was examined in the study. Here, an observation specific to cationic polymer, polyDADMAC, however, surface properties can be quantified  to evaluate a surface characterization feature that can be made in other cationic polymers. What is the positive charge density/cm2 and surface potential? The value can be evaluated by such a techniques using NaFluorecsein, zeta potential etc.  This will also help us  to examine for other interpolyelectrolyte complexes for the relationship between biocidal activity and thresold concentration of overall charge density​​.